# The Interleukin-11/IL-11 Receptor Promotes Glioblastoma Survival and Invasion under Glucose-Starved Conditions through Enhanced Glutaminolysis

**DOI:** 10.3390/ijms24043356

**Published:** 2023-02-08

**Authors:** Sarah F. Stuart, Ayenachew Bezawork-Geleta, Zammam Areeb, Juliana Gomez, Vanessa Tsui, Ahmad Zulkifli, Lucia Paradiso, Jordan Jones, Hong P. T. Nguyen, Tracy L. Putoczki, Paul V. Licciardi, George Kannourakis, Andrew P. Morokoff, Adrian A. Achuthan, Rodney B. Luwor

**Affiliations:** 1Department of Surgery, The Royal Melbourne Hospital, The University of Melbourne, Parkville, VIC 3050, Australia; 2Fiona Elsey Cancer Research Institute, Ballarat, VIC 3350, Australia; 3Department of Physiology, The University of Melbourne, Melbourne, VIC 3010, Australia; 4Department of Neurosurgery, The Royal Melbourne Hospital, Parkville, VIC 3050, Australia; 5The Walter and Eliza Hall Institute of Medical Research, Parkville, VIC 3050, Australia; 6Department of Medical Biology, The University of Melbourne, Parkville, VIC 3050, Australia; 7Murdoch Children’s Research Institute, Melbourne, VIC 3052, Australia; 8Department of Paediatrics, The University of Melbourne, Melbourne, VIC 3052, Australia; 9Health, Innovation and Transformation Centre, Federation University, Ballarat, VIC 3350, Australia; 10Department of Medicine, The Royal Melbourne Hospital, The University of Melbourne, Parkville, VIC 3050, Australia

**Keywords:** IL-11, glioblastoma, glutaminolysis, survival, invasion

## Abstract

Glioblastoma cells adapt to changes in glucose availability through metabolic plasticity allowing for cell survival and continued progression in low-glucose concentrations. However, the regulatory cytokine networks that govern the ability to survive in glucose-starved conditions are not fully defined. In the present study, we define a critical role for the IL-11/IL-11Rα signalling axis in glioblastoma survival, proliferation and invasion when cells are starved of glucose. We identified enhanced IL-11/IL-11Rα expression correlated with reduced overall survival in glioblastoma patients. Glioblastoma cell lines over-expressing IL-11Rα displayed greater survival, proliferation, migration and invasion in glucose-free conditions compared to their low-IL-11Rα-expressing counterparts, while knockdown of IL-11Rα reversed these pro-tumorigenic characteristics. In addition, these IL-11Rα-over-expressing cells displayed enhanced glutamine oxidation and glutamate production compared to their low-IL-11Rα-expressing counterparts, while knockdown of IL-11Rα or the pharmacological inhibition of several members of the glutaminolysis pathway resulted in reduced survival (enhanced apoptosis) and reduced migration and invasion. Furthermore, IL-11Rα expression in glioblastoma patient samples correlated with enhanced gene expression of the glutaminolysis pathway genes GLUD1, GSS and c-Myc. Overall, our study identified that the IL-11/IL-11Rα pathway promotes glioblastoma cell survival and enhances cell migration and invasion in environments of glucose starvation via glutaminolysis.

## 1. Introduction

Glioblastoma is the most aggressive and lethal brain tumor in adults in part due to the sustained proliferative, invasive and pro-survival attributes of cancer cells [1]. Support of these pro-tumorigenic features requires persistent uptake of several nutrients including glucose and the amino acid glutamine that support the substrate synthesis required for critical cellular function and survival [2]. Glucose enters the cell through glucose transport (GLUT) receptors and is metabolized to pyruvate, which is then converted to lactate or acetyl-CoA before entering the tricarboxylic acid (TCA) cycle, which coordinates energy production and biosynthesis [3,4]. Since glucose deprivation is common in the glioblastoma microenvironment [5], glioblastoma cells must adapt to low-glucose concentrations to survive. [6,7,8]. It is well established that glioblastoma cells adapt to metabolic plasticity utilising several mechanisms to survive in periods of glucose shortage [9,10,11]. Since the brain microenvironment is rich in glutamine [2], glioblastoma cells can take advantage of glutamine catabolism (termed glutaminolysis) as an additional or alternative energy source especially when glycolytic energy production is low due to phases when glucose levels are depleted [12].

Glutamine can act as a precursor for protein, nucleotide, fatty acid biosynthesis, redox balance and nicotine adenine dinucleotide (NADH) production [13,14]. It enters the cell mainly through alanine-serine-cysteine transporter 2 (ASCT2; also known as SLC1A5) [15,16] and is converted to glutamate by glutaminase (GLS) [17,18]. Glutamate can then be converted to α-ketoglutarate (αKG) by glutamate dehydrogenase (GLUD1/GDH1) which can then trigger NADH generation and the TCA cycle, promoting the oxidative phosphorylation (OXPHOS) pathway or the reductive carboxylation pathway [19,20]. Thus, glucose and glutamine can compensate for each other to maintain TCA cycle function, promoting cell survival [11]. In addition, glutamine and glutamate can contribute to non-essential amino acid synthesis and be converted to glutathione by enzymes including glutathione synthetase (GSS), leading to maintained redox homeostasis and reduced apoptosis [21].

Metabolic shift or enhanced glutamine metabolism is believed to occur in response to oncogenes such as c-MYC and pro-inflammatory cytokines in glioblastoma [22] However, the exact mechanism by which this occurs is not fully understood in glioblastoma [23]. A major component of the glioblastoma microenvironment milieu is a variety of cytokines and growth factors, including the members of the IL-6 family, which collectively initiate and mediate a range of cellular activities essential to tumor growth [24,25,26]. IL-11, a member of the IL-6 cytokine superfamily, is a pleiotropic cytokine that binds to its specific receptor (IL-11Rα) and the transmembrane co-receptor gp130 [27,28]. Formation of this heterotrimer results in the activation of JAK proteins and subsequent phosphorylation and activation of the transcription factor STAT3, leading to enhanced transcription of many pro-tumorigenic genes [29,30,31]. IL-11 has been identified as a driver of pro-tumorigenic signals in a wide range of malignancies including breast, prostate, endometrial, ovarian, liver and gastrointestinal cancers and is considered an important biomarker in determining the prognosis of patients [25,30,32,33,34,35]. The contribution of IL-11/IL-11Rα signalling to glioblastoma progression is under-explored. We found that IL-11 expression correlates with glioblastoma patient survival. We demonstrate that under conditions of glucose starvation, IL-11Rα enables glioblastoma cells to utilise glutamine for survival, an adaption that protects glioblastoma cells from induced apoptosis. Our data establish a novel IL-11Rα signalling–glutaminolysis metabolism axis where IL-11/IL-11Rα can promote glutamine metabolism and subsequently enhance survival of glioblastoma cells in low-glucose microenvironments. Our results suggest that targeting IL-11 signalling is a potential mechanism to overcome this adaption mechanism.

## 2. Results

### 2.1. IL-11/IL-11Rα Expression Is Elevated in Glioblastoma Tumors and Primary Cell Lines

Since the contribution of the IL-6 family of cytokines to glioblastoma progression has not previously been explored we first determined whether the expression of individual members of this cytokine family correlated with glioblastoma patient survival using the publicly available TCGA datasets. Data were available for six cytokines in the IL-6 family in the TCGA database, with only IL-11 expression significantly correlated with poor glioblastoma patient survival (Figure 1A; Appendix A). We also observed that the expression of IL-11 was significantly higher in glioblastoma compared to low-grade glioma samples (Appendix A), suggesting that IL-11 may play a role in the more aggressive glioma grades. This was consistent with analysis of our glioblastoma patient tissue samples which revealed that IL-11 protein expression correlated with poor survival (Table 1).

We also found that the gene expression levels of IL-11 and IL-11Rα were increased in three of five primary glioblastoma cell lines and those patients in which the cell lines originated had decreased survival (Figure 1B; Appendix A). Consistent with this, SOCS3, an Il-11-STAT3 target gene, was only elevated in the three primary cell lines with high IL-11Rα expression and not in the two cell lines with low IL-11Rα expression (Figure 1C). To further establish a model system to study the role of IL-11 signalling in glioblastoma cells, we stably transfected an IL-11Rα construct into the two cell lines (#20 and #28) with the lowest endogenous IL-11Rα expression. These cell lines were designated #20-IL-11Rα and #28-IL-11Rα (Figure 1D).

### 2.2. IL-11Rα Expression Promotes Cancer Cell Survival in Glucose-Starved Conditions

As glioblastoma cells are often in microenvironments with low glucose concentrations [5], we first examined whether #20-IL-11Rα and #28-IL-11Rα cells displayed differential metabolic properties to their parental counterparts. Both parental and IL-11Rα-transfected matched cells displayed similar ^14^C-glucose oxidation (Figure 2A); however, the #20-IL-11Rα and #28-IL-11Rα cells exhibited significantly greater ^14^C-glutamine oxidation (Figure 2B) and contained greater levels of the direct derivative of glutamine, glutamate, compared to their parental counterparts (Figure 2C). Thus, we hypothesised that IL-11Rα may drive glutaminolysis and subsequently provide enhanced pro-tumorigenic properties. To test this notion, we cultured our parental and IL-11Rα transfected cells in glucose-free media in the presence or absence of glutamine. Both #20-IL-11Rα and #28-IL-11Rα cells demonstrated significantly greater survival in glucose-free media containing glutamine compared to control cells (Figure 2D), but minimal cell survival was observed in both the parental and the IL-11Rα-transfected cells when cultured in glucose- and glutamine-free media. Similarly, when cells were cultured in RPMI glucose-free media containing glutamine, both #20-IL-11Rα and #28-IL-11Rα cells demonstrated significantly greater survival compared to the control cells (Appendix A). This was reflected by the percentage of apoptotic cells (indicated by active caspase 3/7), which was significantly higher in the parental cells compared to the IL-11Rα transfected cells when cultured in glucose-free, glutamine-containing media (Figure 2E). Consistent with this observation, we observed that the expression of the anti-apoptotic molecule Bcl-2 was significantly elevated in the IL-11Rα-transfected cells (Appendix A), which may contribute to the reduction in apoptosis. Glioblastoma patient samples from our cohort and the TCGA also displayed a correlation between higher IL-11Rα and Bcl-2 gene expression (Appendix A). We also observed that both #20-IL-11Rα and #28-IL-11Rα cells demonstrated significantly greater survival compared to control cells when treated with either the non-metabolisable glucose competitive analogue 2-DG or Lopinavir, an inhibitor reported to block glucose transport into the cell in glucose-replete media (Figure 2F,G). To validate that this enhanced survival was due to IL-11Rα, we knocked down IL-11Rα or inhibited IL-11Rα signalling using two inhibitors previously shown to block IL-11Rα signalling, Ponatinib and Bazedoxifene [36,37]. Knockdown of IL-11Rα (Figure 2H) or inhibition with either Ponatinib (Figure 2I) or Bazedoxifene (Figure 2J) led to reduced survival of #20-IL-11Rα and #28-IL-11Rα cells in glucose-free, glutamine-containing media. Similarly, Ponatinib or Bazedoxifene treatment led to significantly greater apoptosis of #20-IL-11Rα and #28-IL-11Rα cells in glucose-free media (Figure 2K).

### 2.3. IL-11Rα Expression Promotes Survival through Glutaminolysis

As both #20-IL-11Rα and #28-IL-11Rα cells displayed greater survival in glucose-free but glutamine-containing media and could oxidize greater levels of glutamine than their parental counterparts we further explored the utilization of glutaminolysis as a source of survival. We cultured the #20-IL-11Rα and #28-IL-11Rα cells in glucose- and glutamine-free media with supplementation of three critical molecules involved in glutaminolysis: L-glutamine, L-glutamic acid (the neutral form of glutamate) or dimethyl 2-oxoglutarate (2-MOG; analogue of αKG). Each of these supplements enhanced #20-IL-11Rα and #28-IL-11Rα cells survival (Figure 3A) and significantly reduced apoptosis (Figure 3B) in glucose- and glutamine-free media. Subsequently, treatment of #20-IL-11Rα and #28-IL-11Rα cells with inhibitors of enzymes involved in the progression of several steps of glutaminolysis led to reversal of survival of the #20-IL-11Rα and #28-IL-11Rα cells in glucose-free, glutamine-containing media. These inhibitors included BenSer (ASCT2 inhibitor; Figure 3C (i)), DON (glutamine antagonist; Figure 3C (ii)), BPTES and CB-839 (GLS inhibitors; Figure 3C (iii,iv)), EGCG and R162 (GLUD1 inhibitors; Figure 3C (v,vi)) and BSO (GSS inhibitor; Figure 3C (vii)). Similarly, these inhibitors also significantly promoted apoptosis in glucose-free, glutamine-containing media (Figure 3D) compared to vehicle-treated cells.

### 2.4. IL-11Rα-Driven Migration and Invasion Is Glutamine-Dependent but Glucose-Independent

As #20-IL-11Rα and #28-IL-11Rα cells demonstrated enhanced survival in glucose-free, glutamine-containing media compared to parental controls, we next determined whether these cells also displayed differential migratory and invasive capabilities in the absence of glucose and/or glutamine. The degree of #20-IL-11Rα and #28-IL-11Rα migration cultured in media containing both glucose and glutamine was similar to that of cells cultured in glucose-free, glutamine-containing media (Figure 4A). However, the migration of #20-IL-11Rα and #28-IL-11Rα cells was significantly reduced when cultured in media containing glucose without glutamine compared to media containing both glucose and glutamine (Figure 4B). Scratch assays also showed that wound closure occurred at a similar rate in media contain glutamine with or without glucose. (Figure 4C; Appendix A). Likewise, #20-IL-11Rα and #28-IL-11Rα cell invasion was not affected by the absence of glucose but was significantly reduced in the absence of glutamine (Figure 4D,E).

### 2.5. IL-11Rα Expression Correlates with Glutamine–Glutamate-Related Genes in Glioblastoma Patient Samples

As we have shown that IL-11Rα allows for glutaminolysis, glutaminolysis-dependent survival, and reduced apoptosis in glucose-starved conditions, and as we observed that inhibition of GLUD1 by EGCG and R162 and GSS by BSO (two key enzymes of glutaminolysis) led to the reversal of this survival, we next determined if IL-11Rα expression correlated with GLUD1 and GSS expression in glioblastoma patient samples. To evaluate correlations, glioblastoma patients were stratified into a low-IL-11Rα-expressing group and a high-IL-11Rα-expressing group. Indeed, glioblastoma patient tumor tissue with low IL-11Rα gene expression also displayed low levels of GLUD1 and GSS, while patient tumor tissue containing high IL-11Rα gene expression also contained significantly higher levels of GLUD1 and GSS gene expression (Figure 5A,B). Likewise, GLUD1 and GSS gene expression was also significantly higher in the #20-IL-11Rα and #28-IL-11Rα cell lines compared to their parental counterparts (Figure 5C). As c-MYC has been shown to promote glutaminolysis in glioblastoma cells [22], we explored whether IL-11Rα expression correlated with c-MYC expression. Indeed, patient tumor samples with low IL-11Rα expression also displayed low levels of c-MYC gene expression while samples with high IL-11Rα expression correlated with levels of c-MYC gene expression (Figure 5D). Consistently, c-MYC gene expression was also significantly higher in the #20-IL-11Rα and #28-IL-11Rα cell lines compared to their parental counterparts (Figure 5E). This suggests that IL-11Rα may be inducing c-MYC expression which then may allow for glutaminolysis and enhanced survival of glioblastoma cells in glucose-depleted conditions.

## 3. Discussion

The human brain is typically a nutrient-enriched environment and utilizes glucose as a major energy source [38]. Glioblastoma cells, however, are often presented with hostile and dynamic conditions such as rapid depletion of nutrients (e.g., glucose) and low oxygen availability in the tumor microenvironment [5]. Therefore, glioblastoma cells must adapt to these sub-optimal conditions using alternative metabolic pathways or succumb to cell death/apoptosis [39,40]. Indeed, glutamine and glutamate levels are often elevated in glioblastoma patient samples indicating an adaption to glutamine metabolism [41]. However, further investigation into the key molecular mechanisms that drive glutaminolysis in glioblastoma have not been completely explored.

Our data proposes a novel mechanism for metabolic adaption, identifying a regulatory link between cytokine signalling and glutaminolysis (Figure 6).

Here, we specifically demonstrate that IL-11Rα promotes the utilization of glioblastoma cells in glucose-deprived conditions to a glutamine-dependent metabolic mechanism subsequently providing protection from apoptosis mediated by glucose starvation. Pharmacological impairment or knockdown of IL-11Rα reduced glutamine metabolism and reduced cell survival in glucose-starved conditions. This novel mechanism of IL-11 signalling may partially explain why glioblastoma patients with higher levels of IL-11 have poorer overall survival. Interestingly, IL-11 expression was the only cytokine in the IL-6 family to correlate to glioblastoma patient outcomes. We propose that high IL-11/IL-11Rα expression leads to reduced apoptosis and subsequently greater glioblastoma progression and poorer patient survival rates potentially through enhanced glutaminolysis. Indeed, our data also demonstrated that IL-11Rα expression positively correlated with the expression of the anti-apoptotic protein Bcl-2 in both clinical samples and cell lines and that high IL-11Rα inhibited caspase-3 activity in glucose-free environments.

High expression of IL-11Rα in our cells resulted in enhanced glutamine oxidation and glutamate levels. Furthermore, high IL-11Rα expression also correlated with increased expression of GLUD1 and GSS, two key enzymes involved in glutaminolysis in both our glioblastoma cell lines and patient tumor tissue. In addition, targeting these enzymes with several inhibitors resulted in a reversal of IL-11Rα-driven survival in glucose-starved conditions. Thus, our data support the notion that IL-11Rα regulates the expression of key enzymes that drive glutamine metabolic processes and sustains cell survival in glucose-poor conditions. Similarly, Yang et al., demonstrated that EGFR activation could drive expression of GLUD1 in glioblastoma cell lines [42]. However, they did not explore whether this enhanced GLUD1 expression resulted in greater survival in the absence of glucose and/or glutamine. Similarly, IL-11Rα expression also correlated to enhanced GSS expression, an enzyme required to produce glutathione, leading to maintained redox homeostasis and reduced apoptosis [21]. Rapidly proliferating cells such as glioblastoma cells produce high levels of reactive oxygen species (ROS), which leads to increased oxidative stress. Glutathione prevents ROS-mediated cell death and therefore promotes tumor-cell survival [43]. Our results expand on this notion, suggesting that IL-11Rα drives enhanced GSS expression and subsequent glutathione production, leading to reduced apoptosis triggered by glucose starvation.

Both glutamine and glutamate have been identified as prognostic biomarkers of glioblastoma [3]. Consistently, glutamine was required for survival of our glioblastoma cell lines even in the presence of glucose. This is in agreement with Wise and colleagues who demonstrated that glutamine was required for the maintenance of glioma cell viability [22]. Zhang et al. also reported that glutamine depletion resulted in glioblastoma cell death through caspase-3 activation [44]. Although it did not induce apoptosis, glutamine starvation hindered the cell proliferation of several glioblastoma cell lines [21,45]. We demonstrated that glutamine was also required for wound healing, transwell migration and invasion in our IL-11Rα-transfected cells. However, neither migration nor invasion of our cells required glucose supplementation. This is consistent with previous findings that have found that glutamine is more important for these cellular processes, albeit our data are the first to demonstrate glutamine-dependent migration and invasion in the glioblastoma setting. In a study conducted by Chen and colleagues, inhibition of glutamine reduced the ability of thyroid cancer cells to both migrate and invade [46]. Furthermore, in a study conducted by Yang and colleagues, glutamine, but not glucose, was found to regulate the STAT3-dependent invasive potential of ovarian cancer cells [47]. This study also established a correlation between increased glutaminolysis gene expression and decreased patient survival in ovarian cancer, while glycolytic genes correlated with a better prognosis [47]. Therefore, collectively our data and those of others suggest that targeting glucose metabolism alone may not yield optimal outcomes due to the capability of some cancers, such as glioblastoma, to switch to a glutamine-dependent metabolic survival. Indeed, a phase II clinical trial testing the efficacy of the GLUT inhibitors Lopinavir and Ritonavir only produced a complete response to treatment in one recurrent high-grade glioma patient, while 79% experienced progression of the disease [48]. We subsequently propose that targeting glutaminolysis may offer an improved treatment strategy for glioblastoma patients.

MYC is a pro-tumorigenic gene upregulated in many types of cancers, including glioblastoma [49]. High MYC expression is required for glutaminolysis and addiction to glutamine metabolism [22] through several mechanisms, including increasing GLUD1 activity [50,51,52]. Additionally, IL-11 has been shown to upregulate the oncogene c-MYC [53,54]. In our present study, we connect these independent findings identifying that IL-11Rα expression correlates with both increased c-MYC and GLUD1 expression in glioblastoma patient tumor tissue and enhanced glutaminolysis leading to significantly greater survival in glucose-starved conditions. Our study supports the report by Le et al., who showed that Burkitt lymphoma cells with high MYC expression were able to survive in the absence of glucose provided these cells were cultured in the presence of glutamine as an alternative energy source [55].

It should be noted that our study focused on primary glioblastoma and not secondary glioblastoma patient tissue and patient-derived cell lines. As 96% of patients with primary glioblastoma contain wt IDH1 (as compared to only 27% of secondary glioblastoma) [56], our study is only relevant to potential IL-11Rα-enhanced survival advantages in the primary glioblastoma context. Interestingly, a recent study suggests that tumors with IDH1 mutations cannot consume glucose as well as cells with wt IDH1, [57], and thus we could speculate that IL-11 signaling and enhanced glutamine metabolism used for survival in secondary glioblastoma may be even more pronounced due to their reduced ability to consume glucose. Investigation into IL-11Rα signaling, glutaminolysis and cell survival in regard to primary versus secondary glioblastoma is thus worth pursuing. A limitation of our study is that we did not determine the exact mutational status of several common genes in glioblastoma, including IDH, EGFR (or the presence of the common EGFR variant, EGFRvIII), MGMT, TP53, RB, PTEN, NF1, MDM2 and loss of chromosome arm 10q in our glioblastoma patient tumor samples and patient-derived cell lines, as this was beyond the scope of our current study. Nonetheless, in summary, our data suggest that IL-11Rα contributes to the ability of glioblastoma cells to metabolically adapt and subsequently survive in glucose-deprived conditions. Specifically, our findings suggest that IL-11Rα promotes increased glutaminolysis and survival in periods of low glucose concentrations in the microenvironment, potentially through a novel mechanism involving increased c-MYC GLUD1 and GSS expression. Targeting this pathway as shown in our study could prevent the adaptive survival of glioblastoma cells in glucose-starved conditions and thus promote cell death in this and potentially other tumor types.

## 4. Materials and Methods

### 4.1. Patient Samples

Formalin-fixed paraffin-embedded sections of glioblastoma tumor tissue was obtained from pathologically confirmed glioblastoma patients at the Royal Melbourne Hospital, Melbourne, VIC, Australia. The available clinical information and treatment of these primary glioblastoma patients are outlined in Appendix A. Use of these human glioblastoma tumor tissues in the laboratory for both immunohistochemistry and gene expression analysis was approved by the Melbourne Health Human Research and Ethics Committee (HREC 2012.136). Approved 25 October 2012.

### 4.2. Inhibitors and Reagents

Ponatinib, Bazedoxifene and Lopinavir were purchased from Selleck Chemicals (Houston, TX, USA). 2-Deoxy-D-glucose (2-DG), O-benzyl-L-Serine (BenSer), 6-diazo-5-oxo-norleucine (DON), bis-2-(5-phenylacetamido-1,2,4-thiadiazol-2-yl)ethyl sulfide (BPTES), Telaglenastat (CB-839), epigallocatechin gallate (EGCG), 2-allyl-1-hydroxy-9,10-anthraquinone (R162), L-buthionine-sulfoximine (BSO), L-glutamine, glutamic acid and dimethyl 2-oxoglutarate were all purchased from Sigma (Sigma-Aldrich, St. Louis, MO, USA). The caspase-3 fluorescence dye was from Sartorius (Sartorius AG, Göttingen, Germany). Human IL-11Rα and negative control siRNA were from Thermofisher Scientific (Scoresby, VIC, Australia).

### 4.3. Cell Culture

The primary glioblastoma cell lines, #15, #20, #28, #35 and #41, were originally derived from 5 patients with pathologically confirmed glioblastoma at the Royal Melbourne Hospital and subsequently modified from neurosphere non-adherent cells to adherent cells grown in monolayer by disassociating spheroid cultures and seeding cells onto adhere plates. Use of these cell lines in the laboratory was approved by the Melbourne Health Human Research and Ethics Committee (HREC 2012.219). Approved 10 December 2012. All cell lines were maintained in Dulbecco’s Modified Eagle’s Medium (Life Technologies, Carlsbad, CA, USA) that contained 5% (*v*/*v*) foetal bovine serum (FBS) (Life Technologies), 100 U/mL penicillin and 100 µg/mL streptomycin (Life Technologies). Transient transfection was performed using Metafectene Pro (Biontex; München, Germany), as per the manufacturer instructions with control or IL-11Rα siRNA. The #20 and #28 IL-11Rα stably transfected clones were generated by transfecting cells with an IL-11Rα construct (R&D Systems, Minneapolis, MN, USA) using FuGENE HD transfection reagent (Promega, Madison, WI, USA) following the manufacturer’s instructions and selecting with Geneticin (Sigma Aldrich, St. Louis, MO, USA). All cells were incubated in a humidified atmosphere of 90% air and 5% CO_2_ at 37 °C. All media with variations of glucose and glutamine concentrations were purchased from Life Technologies.

### 4.4. Cell Viability Assays

Cells were seeded in 24-well plates and allowed to adhere overnight. Triplicate wells were treated with appropriate controls, inhibitors, metabolic analogues, conditioned media and/or glucose and/or glutamine-free media for 3 days. After the treatment period, cells were washed and a mixture of 6.0% glutaraldehyde and 0.5% crystal violet was added for 30 min, followed by another wash, and then allowed to dry overnight. The colonies were quantified using ImageJ Plugin [58].

### 4.5. Immunohistochemistry

Slides were deparaffinised in 100% xylene and rehydrated, after which they were blocked in 5% (*v*/*v*) goat serum followed by an endogenous peroxidases block (Envision™, DAKO; North Sydney, NSW, Australia). Slides were then washed in TBST followed by immunostaining with IL-11 ab (1:25 dilution; Santa Cruz, Dallas, TX, USA) overnight at 4 °C. Sections were subsequently incubated with an anti-rabbit HRP-labelled polymer (Envision™, DAKO) as per the manufacturer’s instructions and then washed in TBST. DAB (Envision™, DAKO) was then added on the sections for 5 min at room temperature followed by immediate immersion in distilled water. Slides were then stained with haematoxylin for 15 s and placed in Scott’s tap water for 15 s. Following dehydration, slides were then mounted with DPX mounting media onto a coverslip and analysed using a Leica DM2000 microscope (Leica Microsystems; North Ryde, NSW, Australia). The sections were observed under a microscope at 200× magnification and scored for absence of staining or positive staining. The staining intensity was then correlated with patient survival. *p* < 0.05 indicates statistical significance.

### 4.6. RNA Extraction and RT-PCR

Cells were seeded in 6-well plates and allowed to adhere overnight. Following cell treatments and/or transfections, total RNA was extracted using an RNeasy Mini Kit (Qiagen; Hilden, Germany) following the manufacturer’s instructions. Similarly, RNA was first extracted from glioblastoma samples on formalin-fixed, paraffin-embedded (FFPE) slides using a PureLink FFPE Total RNA Extraction Kit (Invitrogen, cat# KI560-02, Waltham, MA, USA) following the manufacturer’s instructions, including performing the DNA digestion step prior to reverse transcription. Reverse transcription was performed using the High-Capacity RNA-to-cDNA Kit (Applied Biosystems; Waltham, MA, USA). Reverse Transcription-PCR was performed using the GeneAmp PCR System 2400 (Perkin Elmer, Waltham, MA, USA) under the conditions of 37 °C for 60 min and 95 °C for 5 min. To quantify the transcripts of the genes of interest, real-time PCR was performed using the ViiA 7 Real-Time PCR system (Applied Biosystems) for IL-11 (Applied Biosystems, Hs01055414_m1), IL-11Rα (Applied Biosystems, Hs00234415_m1), SOCS3 (Applied Biosystems, Hs02330328_s1), Bcl-2 (Applied Biosystems, Hs0060823_m1), GLUD1 (Applied Biosystems, Hs03989560_s1), GSS (Applied Biosystems, Hs00609286_m1), c-Myc (Applied Biosystems Hs00153408_m1) and GAPDH (Applied Biosystems, Hs02758991_g1). Amplified RNA samples were calculated using the 2^−∆∆CT^ method [59].

### 4.7. Scratch/Wound-Healing Assay

Cells were seeded in 6-well plates and grown to 100% confluency, after which a scratch/wound was created on the bottom of each well using a p200 sterile tip. Cells were then cultured in media containing mytomysin C (Sigma) to stop proliferation ± glucose ± glutamine and phase-contrast images were acquired at 0 and 24 h post-scratch. An inverted microscope (IX50 Olympus, Tokyo, Japan) and the Leica Application Suite (LAS v4.5) were used to process and capture images. ImageJ was utilized to quantify wound closure.

### 4.8. Transwell Migration and Invasion Assays

Cells were seeded onto the micropore filter of the top chamber of 24-well transwell plates (Corning, Corning, NY, USA) in the presence or absence of glucose-free and/or glutamine-free media. For invasion assays, the micropore filter was pre-coated with 10% (*v*/*v*) Matrigel 24 h prior. After 24 h, the media was removed from the upper well and cells were incubated in formalin (5 min), crystal violet dye (5 min) and then water (5 min). Cells remaining on the upper side of the micropore filter were removed using a cotton bud and the micropore filter was then mounted on a microscope slide. The slides were imaged at 200× magnification, and the images were analysed using an Image Color Summarizer software to determine the percent of cells that had migrated/invaded through the micropore filter.

### 4.9. Apoptosis Assay

Cells were seeded in a 96-well plate and allowed to adhere overnight. Cells were then cultured in the presence or absence of glucose-containing media, ± glutaminolysis inhibitors or metabolic analogues for 48 h. Incucyte Caspase 3/7 fluorescence dye (Sartorius AG, Göttingen, Germany) was added and 30 min later both cell number (DAPI) and fluorescence (GFP) were imaged under the microscope (10×). Fluorescence-positive cells were determined as positive by eye and the percentage of caspase-positive cells was calculated by determining the number of fluorescent-positive cells X 100 and dividing this number by the total number of cells using DAPI. This was performed across a minimum of 3 random fields for each time point and performed a minimum of n = 3.

### 4.10. Glucose and Glutamine Oxidation Assay

Cells were seeded in 12-well plates and allowed to adhere overnight followed by serum starvation for a further 16 h for a ^14^C-tracer-based metabolism assay. Cells were then “pulsed” in low-glucose DMEM containing 500 μM oleate and 1 μCi/mL of [1-^14^C]-oleate (NEC317050C, PerkinElmer, Waltham, MA, USA) conjugated to 1% (*w*/*v*) BSA for 4 h. Oxidation from glucose and glutamine was measured for 4 h in low-glucose DMEM containing 2 μCi/mL D-[^14^C(U)]-glucose (NET238C001MC; PerkinElmer) or L-[^14^C(U)]-glutamine (NEC451050UC; PerkinElmer). At the completion of the experiment, the culture medium was added to 1 mL 1 M perchloric acid to liberate ^14^CO_2_ derived from oxidation, which was collected in 300 μL 1 M sodium hydroxide and counted on a Tri Carb 2810TR liquid scintillation analyzer (Perkin Elmer, Waltham, MA, USA). Cells were washed 3 times in ice-cold PBS, then scraped with PBS containing 0.1% (*v*/*v*) Triton X-100 and passed through a 27G needle and 1 mL syringe. The cell lysate was added to 2:1 chloroform:methanol (*v*:*v*) and incubated for 2 h with intermittent mixing. The tubes were spun at 200 g for 10 min to separate the upper aqueous phase and lower organic phase. The upper aqueous phase was removed and counted using liquid scintillation to determine incomplete oxidation (i.e., acid-soluble metabolites). The lower organic phase was transferred to a fresh tube, dried under N_2_ at 40 °C, then reconstituted in 2:1 chloroform: methanol and counted with liquid scintillation to assess esterification into complex lipids. Oxidation was calculated as the sum of complete oxidation to ^14^CO_2_ and “incomplete” oxidation. The total uptake was calculated by adding oxidation and incorporation into all cellular lipids. All values were normalized to the total cellular protein (BCA method, ThermoFisher Scientific, Scoresby, VIC, Australia). All data are expressed per mg protein.

### 4.11. Glutamate-Detection Assay

Cells were seeded in a 6-well plate and left to adhere overnight. Cells were then treated with inhibitors or control for 48 h, and then lysed and protein quantified. Glutamate concentration was measured using the Glutamate Assay Kit (Sigma-Aldrich^®^; MAK004) according to the manufacturer’s instructions.

### 4.12. OncoLnc (TCGA)

TCGA gene expression data were obtained using the OncoLnc database (www.oncolnc.org) accessed on 14 February 2021. For a given gene, the gene ID was entered and ‘GBM’ was selected. Patients belonging to either the lower or upper 25th percentiles were chosen for the analysis.

### 4.13. Statistical Analysis

The statistical analyses for all experiments were conducted with unpaired, two-tailed Student’s *t*-tests to assess significance and a minimum threshold of *p* < 0.05 was chosen to determine significance. The survival analyses from OncoLnc used a log-rank *t*-test to determine significance and data were displayed on a Kaplan–Meier plot.

## Figures and Tables

**Figure 1 ijms-24-03356-f001:**
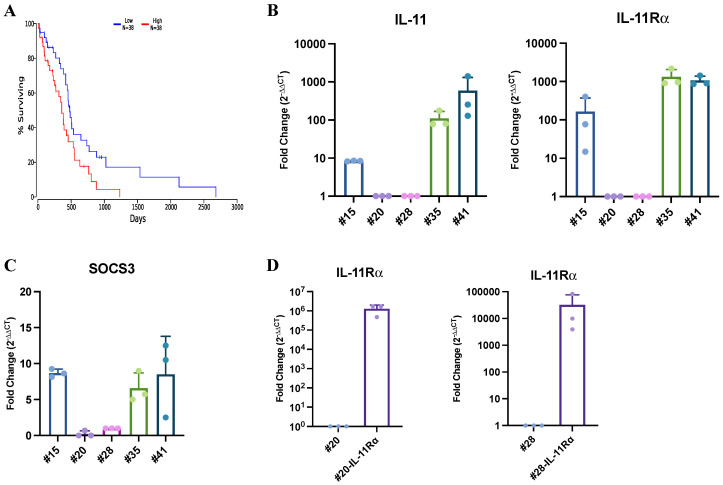
IL-11 expression correlates with poorer survival in glioblastoma patients. (**A**) The relationship between high (red) and low (blue) IL-11 gene expression with patient survival was determined through mining the Oncolnc TCGA dataset. Kaplan–Meier survival curves were evaluated from the TCGA, n = 76; *p* = 0.018. (**B**) Gene expression levels of IL-11 and IL-11Rα and (**C**) IL-11-stimulated SOCS3 were determined in primary glioblastoma cell lines #4, #20, #28, #35 and #41. (**D**) #20, #20-IL-11Rα, #28 and #28-IL-11Rα cells were assessed for IL-11Rα gene expression.

**Figure 2 ijms-24-03356-f002:**
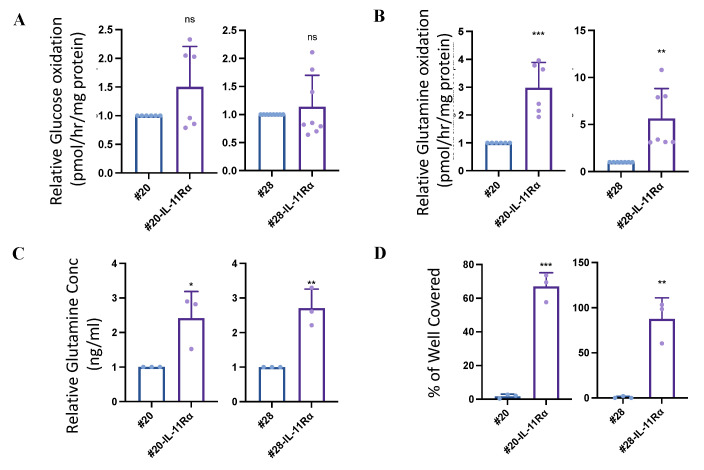
IL-11Rα expression promotes survival in glucose-starved conditions. #20, #20-IL-11Rα, #28 and #28-IL-11Rα cells were serum-starved overnight, then assessed for (**A**) glucose oxidation (n = 6, mean ± SD, where ns indicates *p* > 0.05), (**B**) glutamine oxidation (n = 6, mean ± SD, where ** indicates *p* < 0.01 and *** indicates *p* < 0.001), and (**C**) glutamate production (n = 3, mean ± SD, where * indicates *p* < 0.05 and ** indicates *p* < 0.01). #20, #20-IL-11Rα, #28 and #28-IL-11Rα cells were cultured in glucose-free DME media ± glutamine and evaluated for (**D**) survival using the cell viability assay (n = 3, mean ± SD, where ** indicates *p* < 0.01 and *** indicates *p* < 0.001) and (**E**) percentage of apoptotic cells (active caspase3/7 fluorescence) (n = 3, mean ± SD, where * indicates *p* < 0.05 and ** indicates *p* < 0.01). #20, #20-IL-11Rα, #28 and #28-IL-11Rα cells were cultured in media containing both glucose and glutamine and treated with ± (**F**) 2-DG or (**G**) Lopinavir for 48 h. Survival was evaluated using the cell viability assay (n = 3, mean ± SD, where * indicates *p* < 0.05, ** indicates *p* < 0.01 and *** indicates *p* < 0.001). #20-IL-11Rα and #28-IL-11Rα were (**H**) transfected with control or IL-11Rα siRNA or treated with (**I**) Ponatinib or (**J**) Bazedoxifene and then cultured in glucose-free media containing glutamine for 48 h. Survival was evaluated using the cell viability assay (n = 3, mean ± SD, where ** indicates *p* < 0.01, *** indicates *p* < 0.001 and **** indicates *p* < 0.0001) (**K**) #20-IL-11Rα and #28-IL-11Rα were treated with Ponatinib or Bazedoxifene and then cultured in glucose-free media containing glutamine for 48 h before the percentage of apoptotic cells (active caspase3/7 fluorescence) was determined (n = 3, mean ± SD, where ** indicates *p* < 0.01, *** indicates *p* < 0.001 and **** indicates *p* < 0.0001).

**Figure 3 ijms-24-03356-f003:**
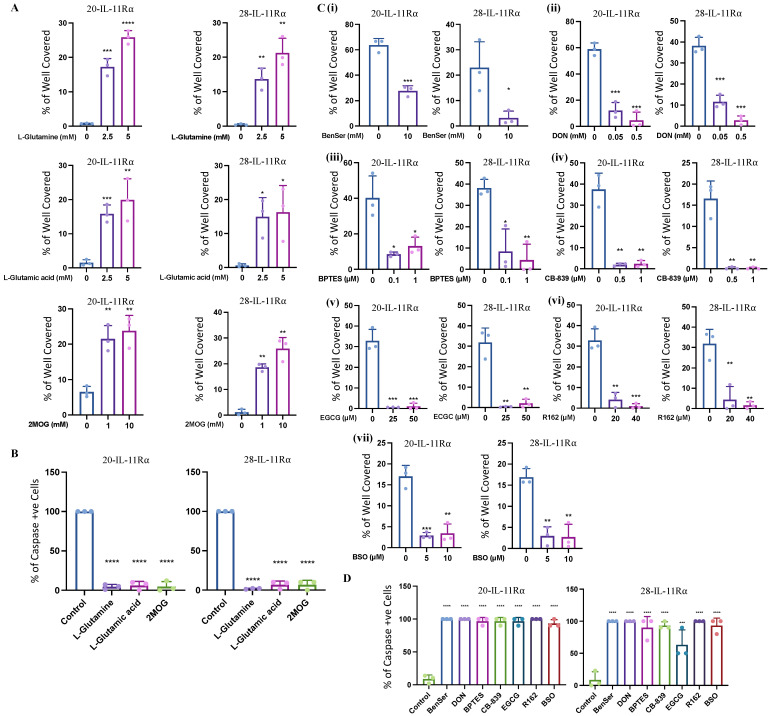
IL-11Rα expression promotes survival through glutaminolysis. #20-IL-11Rα and #28-IL-11Rα cells were cultured in glucose-free and glutamine-free media ± L-glutamine, L-glutamic acid or 2-MOG for 48 h and evaluated for (**A**) survival using the cell viability assay (n = 3, mean ± SD, where * indicates *p* < 0.05, ** indicates *p* < 0.01, *** indicates *p* < 0.001 and **** indicates *p* < 0.0001) and (**B**) percentage of apoptotic cells (active caspase3/7 fluorescence) (n = 3, mean ± SD, where **** indicates *p* < 0.0001). #20-IL-11Rα and #28-IL-11Rα cells were cultured in glucose-free, glutamine-containing media (**C**) (**i**) ± BenSer, (**ii**) DON, (**iii**) BPTES, (**iv**) CB-859, (**v**) EGCG, (**vi**) R162 and (**vii**) BSO for 48 h and evaluated for (**C**) survival using the cell viability assay (n = 3, mean ± SD, where * indicates *p* < 0.05, ** indicates *p* < 0.01, *** indicates *p* < 0.001) and (**D**) percentage of apoptotic cells (active caspase3/7 fluorescence) (n = 3, mean ± SD, where *** indicates *p* < 0.001 and **** indicates *p* < 0.0001).

**Figure 4 ijms-24-03356-f004:**
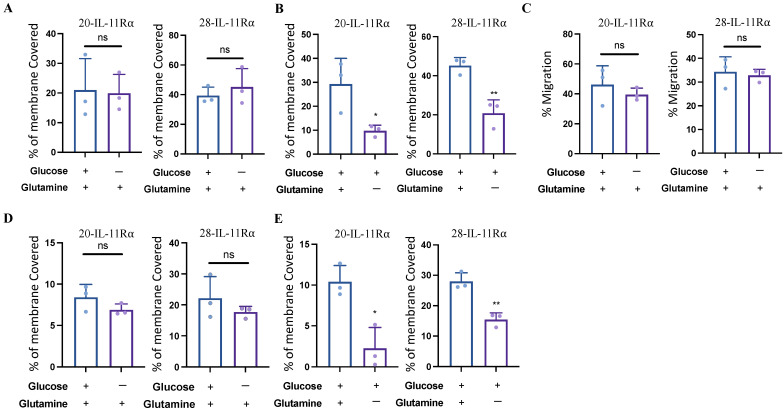
IL-11Rα-driven migration and invasion is glutamine-dependent but glucose-independent. #20-IL-11Rα and #28-IL-11Rα cells were cultured in media ± glucose and ± glutamine as indicated. Cells were then assessed for (**A**,**B**) transwell migration (n = 3, mean ± SD, where ns indicates *p* > 0.05, * indicates *p* < 0.05 and ** indicates *p* < 0.01), (**C**) wound healing (n = 3, mean ± SD, where ns indicates *p* > 0.05) and (**D**,**E**) transwell invasion (n = 3, mean ± SD, where ns indicates *p* > 0.05, * indicates *p* < 0.05 and ** indicates *p* < 0.01).

**Figure 5 ijms-24-03356-f005:**
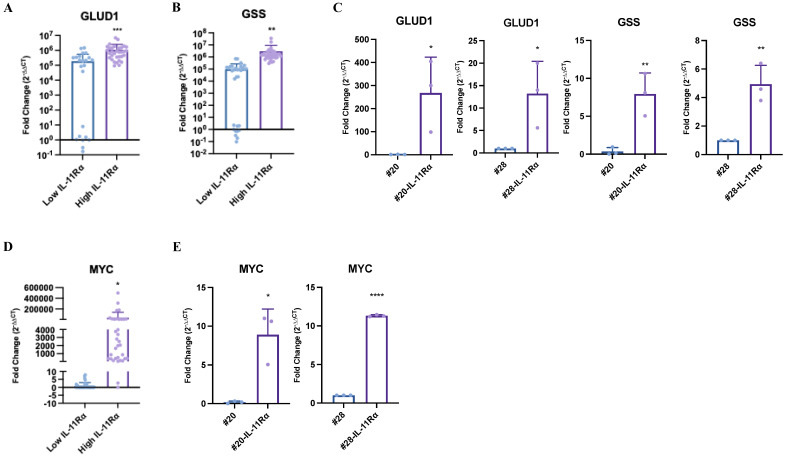
IL-11Rα expression correlates with glutamine–glutamate-related genes in glioblastoma patient samples. Co-expression analysis of IL-11Rα with (**A**) GLUD1 and (**B**) GSS gene expression was performed in glioblastoma patient samples using q-RT-PCR (n = 72, mean ± SD, where ** indicates *p* < 0.01 and *** indicates *p* < 0.001). Expression analysis of (**C**) GLUD1 and GSS gene expression was performed in #20-IL-11Rα and #28-IL-11Rα cells using q-RT-PCR (n = 3, mean ± SD, where * indicates *p* < 0.05 and ** indicates *p* < 0.01). Co-expression analysis of IL-11Rα with (**D**) c-MYC gene expression was performed in glioblastoma patient samples using q-RT-PCR (n = 72, mean ± SD, where * indicates *p* < 0.05). Expression analysis of (**E**) c-MYC gene expression was performed in #20-IL-11Rα and #28-IL-11Rα cells using q-RT-PCR (n = 3, mean ± SD, where * indicates *p* < 0.05 and **** indicates *p* < 0.0001).

**Figure 6 ijms-24-03356-f006:**
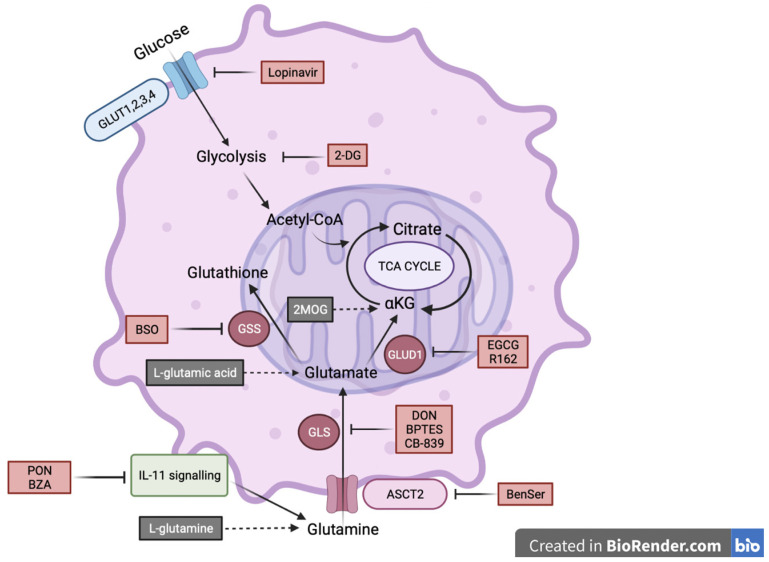
Schematic diagram of IL-11-driven glutaminolysis. IL-11Rα promotes enhanced glutamine oxidation and subsequent glutamate production. In turn, this enhanced glutamate generates increases in glutathione. Lopinavir and 2-DG inhibit glucose oxidation. PON and BZA inhibit IL-11 signalling. BenSer, DON, BPTES, CB-839, EGCG, R162 and BSO inhibit glutamine oxidation. This schematic was created with BioRender.com (accessed on 5 December 2022).

**Table 1 ijms-24-03356-t001:** IL-11 staining with IHC of glioblastoma patient sections.

	<6 Month Survival; n (%)	6–18 Month Survival; n (%)	>18 Month Survival; n (%)
Negative Staining	5 (29)	30 (51)	17 (78)
Positive Staining	12 (71)	29 (49)	5 (22)
Total Samples	17	59	22

## Data Availability

All figures and data within this article can be made available upon request from the corresponding author.

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
