# Peer review of "The Interleukin-11/IL-11 Receptor Promotes Glioblastoma Survival and Invasion under Glucose-Starved Conditions through Enhanced Glutaminolysis"

_ijms, 2023, doi:10.3390/ijms24043356_

Round 1
Reviewer 1 Report
The manuscript is well organized, the study is interesting IL-11/IL-11Rα is identified and correlated with reduced overall survival in glioblastoma 32
patients.
However, the manuscript presents many criticisms:
1- For patient samples the authors should better describe the patients enrolled in the study, age, gender, location of the tumor, type of treatment they underwent; radiotherapy alone, chemotherapy type of chemotherapy, radiotherapy + chemotherapy; they could add a summary table.
2- In the manuscript, the histology of the tumor is mentioned very superficially ... currently the new classification of tumors of the central nervous system 2011 provides for an integrated histological and molecular diagnosis that allows stratifying different biological niches within the same tumor grading.
3- The authors verify their hypothesis in lines stably transfected with IL11Ra, do they have the possibility to verify the hypothesis in wild-type IDH1 glioblastoma lines and IDH1 mutant glioblastoma lines? These cell lines could represent a natural model of glioblastoma with different glucose metabolism.
Author Response
We thank reviewer 1 for their thorough assessment of our manuscript and we welcome their suggested improvements. We have modified our manuscript in line with their comments as outlined below:
Reviewer 1 Comment 1:
For patient samples the authors should better describe the patients enrolled in the study, age, gender, location of the tumor, type of treatment they underwent; radiotherapy alone, chemotherapy type of chemotherapy, radiotherapy + chemotherapy; they could add a summary table.
Our comment to Reviewer 1, comment 1: We thank the reviewer for this important addition. We have included a table outlining all available clinical information regarding the glioblastoma patient used in this manuscript. This information includes gender and treatment received and survival time in days. Unfortunately, age and location of tumor was not made available to our team and therefore we could not include this information. This table now appears in the supplementary material as Suppl Fig 3 and we direct the reader to this information in the methods section, patient samples sub section.
Reviewer 1 Comment 2: In the manuscript, the histology of the tumor is mentioned very superficially ... currently the new classification of tumors of the central nervous system 2011 provides for an integrated histological and molecular diagnosis that allows stratifying different biological niches within the same tumor grading.
Our comment to Reviewer 1, comment 2: We thank the reviewer for there correct assessment of out outlined histo-pathological characterization of our patient tissue. We have stated that the tumors have been pathologically confirmed as glioblastoma in our original manuscript submission. In addition, we have also added that these tumors are all primary glioblastoma and not secondary glioblastoma. This is highlighted in the text (methods section, Patient samples sub section). Unfortunately, we cannot add much more information to our manuscript regarding the sub-classification of these tumors.
Reviewer 1 Comment 3: The authors verify their hypothesis in lines stably transfected with IL11Ra, do they have the possibility to verify the hypothesis in wild-type IDH1 glioblastoma lines and IDH1 mutant glioblastoma lines? These cell lines could represent a natural model of glioblastoma with different glucose metabolism.
Our comment to Reviewer 1, comment 3: This is a very good point raised by reviewer 1. We do not know the IDH1 mutation status of our patient tissue. However, as 96% of primary glioblastoma patients contain wildtype IDH1 we assume that our patients and primary glioblastoma cell lines are wt IDH1. None the less, we have added a paragraph in our discussion regarding this point made by the reviewer. This paragraph states: “It should be noted that our study focused on primary glioblastoma and not secondary glioblastoma patient tissue and patient derived cell lines. As 96% of patients with primary glioblastoma contain wt IDH1 (as compared to only 27% in secondary glioblastoma) [56], our study is only relevant to potential IL-11Rα-enhanced survival advantages in the primary glioblastoma context. Interestingly, a recent study suggests that tumors with IDH1 mutations cannot consume glucose as well as cells with wt IDH1 [57] and thus we could speculate that IL-11 signaling and enhanced glutamine metabolism used for survival in secondary glioblastoma may be even more pronounced due to their reduced ability to consume glucose. Investigation into IL-11Rα signaling, glutaminolysis and cell survival in regard to primary versus secondary glioblastoma is thus worth pursuing.”
Reviewer 2 Report
Stuart and colleagues examined a possible link between expression of interleukin-11 (IL-11)/receptor (IL-11Ra) and survival of patients-derived glioblastoma (GBM) cell lines in glucose-starved conditions. The Authors concluded that cancer cells can survive and migrate in the absence of glucose thanks to utilization of a back-up metabolic pathway - glutaminolysis, which usage is enhanced in cells with the high IL-11/IL-11Ra expression. The study provides an interesting insight into a possible cytokine-driven signaling resulting in a metabolic switch towards glutamine oxidation fostering survival in the stress conditions in GBM cells. Overall study design is fine and minimal applicable methods to support conclusions were employed. Yet, there are some points to be improved which require additional Authors’ attention and consideration.
Major:
· Throughout the manuscript mainly cell lines stably overexpressing IL-11Ra were used (#20 & #28; created from cells characterized by a very low level of endogenous IL-11Ra expression). It is understood that it was done to obtain isogenic pairs of cell lines for further molecular studies, nonetheless the study lacks a control comparison between cell lines characterized by endogenous high levels of IL-11Ra expression and cell lines overexpressing IL-11Ra. Additionally, for overexpression experiments no empty vector control is present. Another possible control would be a knock-down experiment in cells with endogenous high levels of IL-11Ra.
· It is not clear how exactly cells were classified as ‘positive’ for apoptosis fluorescence-based assay. Was it through MFI or fluorescent foci quantification and defined threshold?
· For some experiments 100% of cells is reported as ‘positive’/apoptotic. Is this effect related to prolong exposure to stress conditions (starvation/ inhibitors)? Is there any difference between experimental condition in shorter period (6h or 24h)? Please comment on the rationale for chosen timepoint.
· For better visualization of data exemplary pictures of apoptosis, migration, and scratch assays should be considered as part of main figures.
Minor:
· Descriptions of y-axis in many figures (Figures 2, 3, and 4) are very small and hard to read. Please consider changing the font or layout to more reader friendly.
· The Authors should be applauded for depicting individual values for replicates as single data-points on bar graphs. However, in some cases the individual data points are barely visible (e.g., Figure 2H or Figure 5A,B,C). Please consider using segmented y-axis, logarithmic scale, or log-transformation of data for better visualization.
Author Response
We thank reviewer 2 for their thorough assessment of our manuscript and we welcome their suggested improvements. We have modified our manuscript in line with their comments as outlined below:
Reviewer 2 Comment 1: Throughout the manuscript mainly cell lines stably overexpressing IL-11Ra were used (#20 & #28; created from cells characterized by a very low level of endogenous IL-11Ra expression). It is understood that it was done to obtain isogenic pairs of cell lines for further molecular studies, nonetheless the study lacks a control comparison between cell lines characterized by endogenous high levels of IL-11Ra expression and cell lines overexpressing IL-11Ra. Additionally, for overexpression experiments no empty vector control is present. Another possible control would be a knock-down experiment in cells with endogenous high levels of IL-11Ra.
Our comment to Reviewer 2, comment 1: The reviewer is correct regarding our approach to stably transfect IL-11Rα into two primary glioblastoma cell lines with low endogenous IL-11Rα to obtain isogenic pairs. Unfortunately, we did not create empty vector clones. Nonetheless, this strategy allowed us to compare major cellular and molecular differences between cells with high and low Il-11Rα without other intrinsic, unknown differences between two different cell lines. This is especially important in the glioblastoma setting which is characterized as one of the most heterogenous tumors. Thus, we did not compare differences in the cellular or molecular behavior of varying cell lines based on their IL-11Rα expression status due to the many other unknown molecular differences observed across glioblastoma cell lines.
Knockdown of IL-11Rα in our IL-11Rα over-expressing transfected cell lines showed a reversal of the effects driven by IL-11Rα over-expression. We indeed tried to perform siRNA knockdown of IL-11Rα in 3 cell lines with high endogenous IL-11Rα. However, to our surprise, knockdown of less than 20% of IL-11Rα was achieved despite numerous attempts and slight variations to the protocol. This was in fact performed before we went down the path of knocking down Il-11Rα in the over-expressed cells. Unfortunately, this means we were unable to present any data showing the effects of knocking down IL-11Rα in high endogenously expressing cells in this manuscript.
Reviewer 2 Comment 2: It is not clear how exactly cells were classified as ‘positive’ for apoptosis fluorescence-based assay. Was it through MFI or fluorescent foci quantification and defined threshold?
Our comment to Reviewer 2, comment 2: We thank the reviewer for their great suggestion and apologize for not making this data clearer. We have amended the methods section, apoptosis subsection 4.9 to make this clearer. This section now includes: “The percentage caspase positive cells were calculated by determining the number of fluorescent positive cells x 100 and dividing this number by the number of total cells using DAPI. This was performed across a minimum of 3 random fields for each point and performed a minimum of n = 3 experiments.”
Reviewer 2 Comment 3: For some experiments 100% of cells is reported as ‘positive’/apoptotic. Is this effect related to prolong exposure to stress conditions (starvation/ inhibitors)? Is there any difference between experimental condition in shorter period (6h or 24h)? Please comment on the rationale for chosen timepoint.
Our comment to Reviewer 2, comment 3: The reviewer is correct in stating that the percentage of caspase positive/apoptotic cells is due to reduced glucose levels or the effect of glutaminolysis inhibitors. We also appreciate the question from the reviewer asking if there was any difference in the number of apoptotic cells between IL-11Rα over-expressed and parental cell lines at different time points other than 24 h. We did indeed test this notion evaluating difference at time points ≤ 6 h, 24 h and 48 h. Although cells would have presumably been undergoing the early stages of apoptosis, no cells showed any caspase positivity at ≤ 6 h and thus we did not include these results. The results between 24 h and 48 h were similar with parental cells showing maximal or close to maximal levels of apoptosis (i.e.: 100% caspase positive) at 24 h. Thus, we chose to continue our experiments using this optimized time point of 24 h. All apoptosis data presented in this manuscript is based on 24 h exposure.
Reviewer 2 Comment 4: For better visualization of data exemplary pictures of apoptosis, migration, and scratch assays should be considered as part of main figures.
Our comment to Reviewer 2, comment 4: We appreciate the reviewer’s comment and agree that additional images for apoptosis, migration, invasion and scratch assays would be useful. However, the number of images required to show all controls and media differences and treatment conditions would be overwhelming with each figure requiring at least another 20 images. For example, the apoptosis figure presented in Fig 3D summarizes analysis of 16 different images and thus added these into the original figure we feel would distract the reader with too many figure panels. The figure already contains multiple panels. This is a similar situation to the other main text figures. Overall, we have therefore decided not to include these multiple sets of images to our manuscript.
Reviewer 2 Comment 5: Descriptions of y-axis in many figures (Figures 2, 3, and 4) are very small and hard to read. Please consider changing the font or layout to more reader friendly.
Our comment to Reviewer 2, comment 5: We welcome the reviewer’s comments (both comment 5 and 6) in regard to improving the clarity of the figures. We have changed the axis to make them easier to read and to improve the figure presentation. In some cases, we have used the same Y-Axis label across 2 graphs within the same figure panel. See our improved Fig 2, 3, 4 and Suppl Fig 2.
Reviewer 2 Comment 6: The Authors should be applauded for depicting individual values for replicates as single data-points on bar graphs. However, in some cases the individual data points are barely visible (e.g., Figure 2H or Figure 5A, B, C). Please consider using segmented y-axis, logarithmic scale, or log-transformation of data for better visualization.
Our comment to Reviewer 2, comment 6: Once more we appreciate the reviewer’s comments regarding the presentation of our figures. We have changed the presentation of some of these figure panels (Fig 5) to logarithmic scale to present our data in a more reader-friendly style.
Round 2
Reviewer 1 Report
Considering the clarifications made in the new version of the paper, the authors have improved the presentation of the study. In this form, the manuscript can be considered suitable for publication in the International journal of Molecular Science
Author Response
We thank reviewer 1 for their thorough assessment of our manuscript (twice) and we welcome their suggested improvements. We have modified our manuscript in line with their comments as outlined below:
Reviewer 1:
Reviewer 1, comment 1: Considering the clarifications made in the new version of the paper, the authors have improved the presentation of the study. In this form, the manuscript can be considered suitable for publication in the International journal of Molecular Science
Our response to reviewer 1, comment 1: We thank the reviewer for reading and reviewing our manuscript twice and making valuable suggestions. Based on the reviewer’s comments we did not make any additional changes to the manuscript as they were satisfied with our modifications.
Reviewer 2 Report
I would like to thank the Authors for their thorough answers to my comments and incorporating suggestions in a revised version of the manuscript. Overall changes enhanced the manuscript and made presentation of data and results much clearer. Generally, the manuscript has been improved sufficiently to be accepted for publication. Yet, since the Authors have decided to not include (or could not include) suggested additional experimental data, I would strongly recommend incorporating a “limitation of the study” paragraph in the discussion section containing i.e., Authors’ response to comment #1 and further discussion on molecular heterogeneity of the GBM (e.g., IDH1/2, EGFR, MGMT alterations) which differentiation were beyond experimental span of the study.
Additionally, my question from comment #2 was not about how the percentage of apoptotic cells was calculated, but rather how the fluorescent positive cells were discriminated from the fluorescent negative cells (classified based on MFI threshold, foci quantification, or “read-by-eye”?) – please specify.
Author Response
We thank reviewer 2 for their thorough assessment of our manuscript (twice) and we welcome their suggested improvements. We have modified our manuscript in line with their comments as outlined below:
Reviewer 2 Comment 1: I would like to thank the Authors for their thorough answers to my comments and incorporating suggestions in a revised version of the manuscript. Overall changes enhanced the manuscript and made presentation of data and results much clearer. Generally, the manuscript has been improved sufficiently to be accepted for publication. Yet, since the Authors have decided to not include (or could not include) suggested additional experimental data, I would strongly recommend incorporating a “limitation of the study” paragraph in the discussion section containing i.e., Authors’ response to comment #1 and further discussion on molecular heterogeneity of the GBM (e.g., IDH1/2, EGFR, MGMT alterations) which differentiation were beyond experimental span of the study.
Our response to reviewer 2, comment 1: We thank the reviewer for reading and reviewing our manuscript twice and making valuable suggestions. Based on the reviewer’s comments 1, we have added a limitations paragraph to our discussion section that reads: “A limitation of our study is that we did not determine the exact mutational status of several common genes in glioblastoma including IDH, EGFR (or the presence of the common EGFR variant, EGFRvIII), MGMT, TP53, RB, PTEN, NF1, MDM2 and loss of chromosome arm 10q in our glioblastoma patient tumor samples and patient derived cell lines as this was beyond the scope of our current study.”
Reviewer 2 Comment 2: Additionally, my question from comment #2 was not about how the percentage of apoptotic cells was calculated, but rather how the fluorescent positive cells were discriminated from the fluorescent negative cells (classified based on MFI threshold, foci quantification, or “read-by-eye”?) – please specify.
Our response to reviewer 2, comment 1: We apologise for not interpreting the reviewer’s original comment. We have added further clarification of how we distinguished the fluorescent positive cells from the fluorescent negative cells. This is in the methods section, subsection 4.9 apoptosis assay and includes “Fluoresence positive cells were determined as positive by eye and the percentage of caspase positive cells were calculated by determining the number of fluorescent positive cells X 100 and dividing this number by the number of total cells using DAPI.”